# A novel hybrid PSO-MIDAS model and its application to the U.S. GDP forecast

Feng Shen[1,2], Xiaodong Yan[1], Yuhuang Shang[3]*

1 School of Finance, Southwestern University of Finance and Economics, Chengdu, PR China,
2 Engineering Research Center of Intelligent Finance, Southwestern University of Finance and Economics, Chengdu, PR China, 3 Institute of Chinese Financial Studies, Southwestern University of Finance and Economics, Chengdu, PR China

* syh@swufe.edu.cn

## Abstract

In this study, the traditional lag structure selection method in the Mixed Data Sampling (MIDAS) regression model for forecasting GDP was replaced with a machine learning approach using the particle swarm optimization algorithm (PSO). The introduction of PSO aimed to automatically optimize the MIDAS model's mixed-frequency lag structures, improving forecast accuracy and resolving the "forecast accuracy" and "forecast cost" weighting problem. The Diebold–Mariano test results based on U.S. macroeconomic data show that when the forecast horizon is large, the forecast accuracy of the PSO-MIDAS model is significantly better than other benchmark models. Empirical results show that, compared to the benchmark MIDAS model, the forecast accuracy of both univariate and multivariate PSO-MIDAS models improves by an average of 10% when the forecast horizon exceeds 2 quarters, and the optimization effect is greater compared to other benchmark models. The innovative use of the PSO algorithm addresses the limitations of traditional lag structure selection methods and enhances the predictive potential of the MIDAS model.

## 1. Introduction

Economic growth is one of the ultimate goals of monetary policy. As a crucial macroeconomic fundamental indicator, gross domestic product (GDP) plays a vital role in portraying the economic situation and characterizing the macroeconomic cycle [1]. The future GDP trend is closely related to production planning, consumer decision-making, and policymaking, so it has received extensive attention from producers, consumers, and governments. In the GDP growth forecast, the improvement of forecast accuracy often takes time and cost. Therefore, compared with the forecast timeliness, researchers pay more attention to how to have both "high accuracy" and "low cost".

In specific empirical studies, researchers found that the accuracy of the forecast depends on predictors' choice. The timeliness of forecast is mainly affected by the frequency of data. According to the research of Clements et al. [2, 3] and Ghysels et al. [4], the choice of predictors and the richness of data frequency have a crucial impact on GDP forecast.

To improve the accuracy of forecasting, in the selection of predictors, in addition to traditional macroeconomic variables, financial variables are gradually taken into consideration.

**Data Availability Statement:** All relevant data are within the manuscript and its Supporting Information files.

**Funding:** This study was funded by the National Natural Science Foundation of China (no.

72001178), the project of Research Center for System Sciences and Enterprise Development (no. Xq22B03), the Fundamental Research Funds for the Central Universities (no. JBK2304021, no. JBKZD06005), and the Major Project of Humanities and Social Sciences Key Research Base of the Ministry of Education (no.22JJD790067, no.22JJD790069). The funders had no role in study design, data collection and analysis, decision to publish, or preparation of the manuscript.

**Competing interests:** The authors have declared that no competing interests exist.

Stock et al. [5] and Ghysels et al. [4] are all concerned about financial variables' ability to predict macroeconomic aggregates. Financial variables contain expected information about future economic activities, which is conducive to improving forecasts' accuracy and showing actual availability, bringing about forecast accuracy improvements. Stock et al. [5] pointed out that the predictive power of financial variables for macroeconomic variables is statistically significant. Besides, the predictor can also select other indicators. Lahiri et al. [6] studied the role of the monthly diffusion indices compiled by the Institute for Supply Management (ISM) in predicting the current quarter's GDP growth in the United States. They found new data on the ISM index available at the beginning of the month can improve the real-time forecasting effect.

As far as the richness of data frequency is concerned, high-frequency data has attracted more researchers' attention. For the forecast of low-frequency macroeconomic indicators, high-frequency data information is even more critical. For example, GDP is quarterly data. If we want to introduce high-frequency data to predict low-frequency GDP, we will face a mixed-frequency modeling problem. It is worth noting that Ghysels et al. [7, 8] proposed the MIxed DAta Sampling (MIDAS) regression model, which embeds high-frequency time series lags of the explanatory variables into the regression model through a weighting scheme. The MIDAS model's main advantage is that it can include as much mixed-frequency information as possible and improve forecast accuracy when there are mixed data samples. Moreover, the MIDAS model can use the latest published high-frequency data to improve the timeliness of forecast. The MIDAS model has been proved to be effective in many fields of financial and economic such as financial instrument's price forecasting [9, 10], volatility forecasting [11, 12], and macroeconomic indicator's forecasting [13, 14].

At present, the MIDAS model is widely used in macroeconomic forecasting. Clements et al. [2, 3] and Ghysels et al. [4] apply the MIDAS model to the empirical study of real-time forecast of macroeconomic variables. Kuzin et al. [15] used the MIDAS model to forecast quarterly GDP growth in the euro area in real-time (nowcasting) and forward forecasting and found that, compared with the mixed-frequency VAR model, the MIDAS model performs better when forecasting 4 to 5 months in advance. Furthermore, researchers have made many improvements to the MIDAS model in macroeconomic forecasting. Barsoum et al. [16] combined the unconstrained U-MIDAS model with Markov-switching and proposed a new MS-U-MIDAS model. Empirical research based on U.S. GDP growth data shows that the MS-U-MIDAS model's real-time nowcasting and forward forecasting capabilities are similar to, or better than, traditional MIDAS models. Qiu [17] propose a new tree-based MIDAS model that introduces the regression tree (RT), the bootstrap aggregating decision trees (BAG) and the random forest (RF) algorithms into the MIDAS framework. The empirical results show that the tree-based MIDAS model generally improves forecast accuracy by a wide margin compared to existing MIDAS model in the U.S. Consumer Confidence Index's forecasting. Recently, some researchers have begun to study the optimization of multivariate MIDAS models in price forecasting. Li et al. [18] integrate the extreme learning machine with the multivariate MIDAS model to predict natural gas prices. Wang and Kang [19] combine the multivariate MIDAS model with eXtreme Gradient Boosting (XGBoost) to forecast China's steam coal prices.

Although the MIDAS model provides us with a research paradigm for forecasting economic indicators by mixed-frequency data, we cannot intelligently choose the data richness in the regression of the MIDAS model. The lag order of the predictor determines the amount of information. The greater the lag order, the more lag period data used by the predictor for regression, and the greater the amount of information contained in the model. Generally speaking, the forecast accuracy of the model increases as the richness of information increases.

However, too long lagging information may bring about redundant effects, cause data noise, and ultimately negatively affect the forecast accuracy. Therefore, optimizing the lag structures and improving the MIDAS model's forecast accuracy is a real challenge we face. In the existing research, two traditional methods are generally used to select the maximum lag order: expert experience method and information criterion method. In Clements et al. [3], when the monthly data of predictors are matched with quarterly GDP data, to facilitate comparison with the benchmark model, the high-frequency lags are set to a multiple of 3. Andreou et al. [20] used the AIC criterion to determine the maximum lag order of the MIDAS model in the context of a daily-quarterly data mixture. These two methods are mature in determining the maximum lag order of a single frequency model. However, there are still specific problems in determining the MIDAS model's mixed-frequency lag structures. The incompatibility between the three traditional lag structure selection methods and the MIDAS model are described in Section 2. Given that the above three methods are not suitable for the MIDAS model, the simplest solution is the exhaustive method: traverse all the values of the mixed-frequency lag structure, establish multiple MIDAS models, and finally find the best lag structure. However, the exhaustive method's biggest problem is that the improvement of forecast accuracy comes at the expense of time and cost, "high accuracy" and "low cost" cannot be achieved simultaneously.

In response to this problem, this paper uses machine learning to replace existing mixed-frequency lag structure selection method. We use Particle Swarm Optimization (PSO) to improve the traditional MIDAS model and obtain a PSO-MIDAS model embedded with machine learning algorithms. The main advantages of this model are: on the one hand, it significantly improves the forecast accuracy by optimizing the mixed-frequency lag structures; on the other hand, it cleverly uses the particle swarm optimization algorithm to learn and correct itself during the optimization process, which significantly saves the time and cost. Besides, the PSO-MIDAS model has universal applicability, and different optimal values can be obtained for different data, which improves the applicability of the model.

This paper's main contribution lies in model construction, the machine learning algorithm is embedded in the traditional MIDAS model, and a new PSO-MIDAS model is proposed. We link the lag structure and validation set forecast metric of the MIDAS model to the particle position and fitness function of the PSO algorithm and optimize the lag structure of the MIDAS model using the PSO algorithm. According to economic and financial theory [21], we select the three variables of US industrial production (IP), non-farm payroll (NFP), and capacity utilization (CU) as predictors. This article compares the PSO-MIDAS model with the MIDAS model, U-MIDAS model and ADL model. The study found that the PSO-MIDAS model is improved by an average of 10% relative to the benchmark MIDAS model, and the optimization effect is greater compared to other benchmark models. The Diebold–Mariano test results show that when the forecast horizon is large, the forecast accuracy of the PSO-MIDAS model is significantly better than other benchmark models. This paper also extends the MIDAS model from univariate form to multivariate form. The multivariate model PSO-MV--MIDAS's forecast accuracy is still better than the multivariate benchmark models. In general, the particle swarm optimization algorithm has made a significant contribution to improving the MIDAS model's forecast accuracy.

The rest of this article proceeds as follows. In section 2, we introduce the theories and methods related to this research: the MIDAS model, leads and nowcast, the determination of lag order, PSO algorithm and the PSO-MIDAS model; Section 3 introduces the empirical research design of this article; Section 4 shows the empirical analysis results and discussions; Section 5 concludes the paper.

## 2. Methodology

### 2.1. The MIDAS model

Before introducing the MIDAS model, we need to introduce the Augmented Distributed Lag (ADL) model first. Both the ADL model and the MIDAS model can forecast low-frequency variables through mixed-frequency data. Suppose we hope to forecast some low-frequency quarterly variables through mixed-frequency data in the context of monthly-quarterly data mixtures. The predicted variable, such as the quarterly real GDP growth one quarter ahead, is denoted by $Y_{t+1}^Q$. The predictor variable, such as the monthly industrial production in the first month of the $t$ quarter, is denoted by $X_{1,t}^M$. Note that the ADL model involves temporally aggregated series. We denote the quarterly aggregate of the predictor variable in the $t$ quarter as $X_t^Q$. The aggregation scheme being used, for example, is averaging the monthly variable data, that is, $X_t^Q = (X_{1,t}^M + X_{2,t}^M + X_{3,t}^M)/3$. The ADL($p_Y^Q, q_X^Q$) regression model is as follows:

$$Y_{t+1}^Q = \mu + \sum_{j=0}^{p_Y^Q - 1} \rho_{j+1} Y_{t-j}^Q + \sum_{j=0}^{q_X^Q - 1} \beta_{j+1} X_{t-j}^Q + u_{t+1}^Q, \tag{1}$$

Which includes $p_Y^Q$ lags of $Y_t^Q$ and $q_X^Q$ lags of $X_t^Q$, $\mu$ is the constant and $u_{t+1}^Q$ is the random error term. This regression is parsimonious because only $p_Y^Q + Y_t^Q + 1$ regression coefficients need to be estimated. Unlike the ADL model, the MIDAS model does not need to artificially set up a quarterly aggregation scheme for monthly variable and directly incorporate monthly high-frequency data into the model. Now we introduce the ADL $-$ MIDAS($p_Y^Q, q_X^M$) model with h-steps ahead forecast:

$$Y_{t+h}^Q = \mu^h + \sum_{j=0}^{p_Y^Q - 1} \rho_{j+1}^h Y_{t-j}^Q + \beta^h \sum_{j=0}^{q_X^Q - 1} \sum_{i=0}^{2} w_{i+j*3}^{\theta^h} X_{3-i,t-j}^M + u_{t+h}^Q, \tag{2}$$

Which includes $p_Y^Q$ lags of $Y_t^Q$ and $q_X^M$ lags of $X_{i,t}^M$. For the convenience of notation and explanation, we coincidentally set $q_X^M$ to 3 times $q_X^Q$. However, it is worth noting that the ADL $-$ MIDAS($p_Y^Q, q_X^M$) model can contain monthly high-frequency data of any positive integer lag order $q_X^M$. ADL $-$ MIDAS($p_Y^Q, q_X^M$) assigns different weight coefficients $w_i^{\theta^h}$ to each high-frequency lag. By introducing some weighting schemes to determine the mapping relationship between the low-dimensional hyperparameter $\theta^h$ and the high-dimensional weight coefficient $w_i^{\theta^h}$, the weighting scheme dramatically reduces the number of parameters to be estimated. It avoids the problem of parameter proliferation caused by directly estimating coefficients for each high-frequency lag.

The alternative weighting schemes include U-MIDAS (unrestricted MIDAS polynomial), normalized Beta probability density function, normalized exponential Almon lag polynomial, and polynomial specification with step functions. Ghysels et al. [8] provided a detailed discussion about those weighting schemes. Following Ghysels et al. [8], we adopt the normalized Beta probability density function as the ADL $-$ MIDAS($p_Y^Q, q_X^M$) model weighting scheme, and the weighting scheme formula is as follows:

$$w_i^{\theta^h} = \frac{f(i/q_X^M, \theta_1^h, \theta_2^h)}{\sum_{s=1}^{q_X^M} f(s/q_X^M, \theta_1^h, \theta_2^h)}, \tag{3}$$

with:

$$\theta^h = (\theta_1^h, \theta_2^h) \tag{4}$$

$$f(x, a, b) = \frac{x^{a-1}(1 - x)^{b-1}\Gamma(a + b)}{\Gamma(a) + \Gamma(b)}. \tag{5}$$

Therefore, we do not need to estimate the weight coefficient $w_i^{\theta^h}$ of each high-frequency lag, only need to estimate $\theta_1^h$ and $\theta_2^h$ to determine $w_i^{\theta^h}$. Then the estimated parameter set of the $\mathrm{ADL} - \mathrm{MIDAS}(p_Y^Q, q_X^M)$ model with normalized Beta probability density function weighting scheme is: $(\mu^h, \rho_1^h, \rho_2^h, \ldots, \rho_{p_Y^Q}^h, \beta^h, \theta_1, \theta_2)$, a total of $p_Y^Q + 4$ parameters, which can be estimated by nonlinear least squares (NLS).

In addition, we call the $\mathrm{ADL} - \mathrm{MIDAS}(p_Y^Q, q_X^M)$ model with unrestricted MIDAS polynomial weighting scheme as $\mathrm{U} - \mathrm{MIDAS}(p_Y^Q, q_X^M)$ model. The $\mathrm{U} - \mathrm{MIDAS}(p_Y^Q, q_X^M)$ model is as follows:

$$Y_{t+h}^Q = \mu^h + \sum_{j=0}^{p_Y^Q-1} \rho_{j+1}^h Y_{t-j}^Q + \sum_{j=0}^{q_X^Q-1} \sum_{i=0}^{2} \beta_{i+j*3}^h X_{3-i,t-j}^M + u_{t+h}^Q. \tag{6}$$

The $\mathrm{U} - \mathrm{MIDAS}(p_Y^Q, q_X^M)$ model directly regresses $Y_{t+h}^Q$ with low-frequency data $Y_t^Q$ and high-frequency data $X_{i,t}^M$. The number of parameters to be estimated is $p_Y^Q + q_X^M + 1$, which can be estimated by the ordinary least square (OLS) method. However, when $p_Y^Q$ and $q_X^M$ are large enough, the U-MIDAS model will suffer parameter proliferation issues and reducing forecast accuracy.

## 2.2. Leads and nowcast

In mixed-frequency forecast research, because the publication dates of high-frequency data and low-frequency data are not synchronized, the data we collect is usually ragged-edge data with missing observations at the end of the sample. For example, in a particular calendar month, we can observe the current quarter's monthly data, but we cannot observe the current quarter's quarterly data. So, we need to extend the $\mathrm{ADL} - \mathrm{MIDAS}(p_Y^Q, q_X^M)$ model to include high-frequency monthly data in the current quarter.

We follow the concept based on MIDAS regression with leads proposed by Kuzin et al. [22] and Andreou et al. [18]. When our regression uses the information between quarter $t$ and $t$+1, we call it regression with leads. For example, suppose we are one month into quarter $t$+1, hence the end of January, April, July, or October, our goal is to forecast the quarterly variables in the quarter $t$+$h$. At this time, we will have one-month lead data, denoted by $X_{1,t+1}^M$. Denoted by $L_X^M$ the number of leads, the value range of $L_X^M$ is (0,1,2). Then the extended $\mathrm{ADL} - \mathrm{MIDAS}(p_Y^Q, q_X^M, L_X^M)$ model is as follows:

$$Y_{t+h}^Q = \mu^h + \sum_{j=0}^{p_Y^Q-1} \rho_{j+1}^h Y_{t-j}^Q + \beta^h \Big[ \sum_{i=0}^{L_X^M} w_{-L_X^M}^{\theta^h} X_{L_X^M,t+1}^M + \sum_{j=0}^{q_X^Q-1} \sum_{i=0}^{2} w_{i+j*3}^{\theta^h} X_{3-i,t-j}^M \Big] + u_{t+h}^Q. \tag{7}$$

When $L_X^M = 0$, $\mathrm{ADL} - \mathrm{MIDAS}(p_Y^Q, q_X^M, L_X^M)$ degenerates to $\mathrm{ADL} - \mathrm{MIDAS}(p_Y^Q, q_X^M)$ model. The calculation of the high-frequency lag $q_X^M$ has also changed accordingly, and $q_X^M$ not only includes lags before the quarter t also needs to include the information of the leads $L_X^M$, then $q_X^M = 3*q_X^Q + L_X^M$. In particular, we call the forecast made by the $\mathrm{ADL} - \mathrm{MIDAS}(p_Y^Q, q_X^M, L_X^M)$ model when h = 1 nowcast, that is, forecasting the current quarter's quarterly variables by using the current quarter's monthly data. For more information about nowcast and MIDAS regression with leads, Andreou et al. [18] had already elaborated in detail.

## 2.3. The determination of lags

Before using the $ADL - MIDAS(p_Y^Q, q_X^M, L_X^M)$ model for prediction, we need to determine the model's low-frequency lags $p_Y^Q$ and high-frequency lags $q_X^M$. $p_Y^Q$ and $q_X^M$ directly determine the information richness of the high-frequency and low-frequency data included in the model and determine the number of estimated parameters. Most scholars use the expert experience and information criterion to determine the model's lag structures in the existing research.

Some researchers determine the model's lag structures based on his own research experience or expert advice. It is an empirical method based on a large number of empirical studies. However, determining the model lag structures based on expert experience is highly subjective and prone to model setting errors. Different time series research objects and different data sets will be different in model lag structure choices. Subjective expert experience is challenging to choose a reasonable lag structure on the new model and new data. Therefore, facing more complex MIDAS models, it is impossible to set the model's lag structure through expert experience accurately.

The information criterion is the most commonly used method to determine the model's lag structure. Generally, three information criteria, AIC (Akaike Information Criterion), BIC (Schwarz-Bayesian Information Criterion) or HQC (Hannan–Quinn information criterion), are used to select models. The definitions of AIC, BIC and HQC are:

$$AIC = 2K - 2\log(\hat{L}_{max}), \tag{8}$$

$$BIC = K\log(N) - 2\log(\hat{L}_{max}), \tag{9}$$

$$HQC = 2K\log(\log(N)) - 2\log(\hat{L}_{max}). \tag{10}$$

$K$ is the number of estimated parameters. $N$ is the sample size. $\hat{L}_{max}$ is the maximum value of the likelihood function of the estimated model. We choose the model with the lowest information criterion value as the optimal model by calculating the AIC, BIC or HQC value of the model with different lag structures. It can be seen that the definitions of the three information criteria are relatively similar. Both are composed of the parameter number penalty part and the maximum likelihood function part. The maximum likelihood function part ensures that the best fit of the model within the sample. The parameter number penalty part ensures that the model is as simple as possible and improves its generalization capability out the sample. In the time series model, the lag order is proportional to the number of parameters K, so the information criterion can weigh the fit and generalization effects of the model in and out of the sample and choose the optimal lag order. However, because the information criterion only penalizes the model's total number of parameters, it cannot reflect the model's parameter quantity structure. That is, the information criterion only examines the sum of $p_Y^Q$ and $q_X^M$ and does not examine the structural changes between $p_Y^Q$ and $q_X^M$. Therefore, using the information criterion method to determine the MIDAS model's lag structures has certain theoretical flaws.

Through theoretical analysis, the method of expert experience and information criterion both have specific defects in determining the MIDAS model's lag structures. However, determining $p_Y^Q$ and $q_X^M$ is an essential part of setting the $ADL - MIDAS(p_Y^Q, q_X^M, L_X^M)$ model, which directly affects the forecast capability of the model, so this article proposes another way to determine the lag structures of the MIDAS model. Just as hyperparameters usually need to be adjusted to improve the prediction accuracy of the machine learning model, we regard $p_Y^Q$ and $q_X^M$ as the hyperparameters of $ADL - MIDAS(p_Y^Q, q_X^M, L_X^M)$ model. By comparing the prediction accuracy of each possible model, we can find the optimal combination of $p_Y^Q$ and $q_X^M$. However,

this process will consume a lot of time and computing resources when $p_Y^Q$ and $q_X^M$ are large. For example, when $p_Y^Q$ and $q_X^M \in [1, 10]$, we need to estimate 100 models to find the optimal combination of $p_Y^Q$ and $q_X^M$. However, when $p_Y^Q$ and $q_X^M \in [1, 100]$, the number of models required increases to 10000.

We usually compare the forecast metrics to judge the forecast capability of the model. The root mean squared forecast error (RMSFE) is the square root of the mean squared forecast error, the RMSFE is a measure of the magnitude of a typical forecasting "mistake". The calculation formula of RMSFE is as follow:

$$RMSFE = \sqrt{E[(Y_{t+h} - \hat{Y}_{t+h|t})^2]} \tag{11}$$

$Y_{t+h}$ denotes the true value of the explained variable at period t+h, and $\hat{Y}_{t+h|t}$ denotes the predicted value based on information available at period t. A model with a smaller RMSFE usually has a better forecast capability. We can find that the combination of $p_Y^Q$ and $q_X^M$ has a complicated relationship with the model's RMSFE. Our goal is to minimize the model's RMSFE in the validation set to find the optimal lag structures. As long as we can find a suitable method to solve this minimization problem, we can find the optimal lag structures. Therefore, we introduce a method to solve this optimization problem in the next section.

## 2.4. Particle swarm optimization

Particle Swarm Optimization (PSO) is a self-organizing heuristic optimization algorithm proposed and developed by Kennedy and Eberhart [23, 24]. The invention of this algorithm is inspired by the swarm hunting behavior of birds and fish. It is a kind of swarm intelligent random optimization algorithm. Compared with other heuristic optimization algorithms, particle swarm optimization has high computational efficiency, robust parameter control, and easy implementation and application [25]. Besides, compared to other non-random optimization algorithms, the random strategy of the PSO algorithm allows for improved global optimization capabilities. It does not rely on the optimization problem's strict mathematical properties, so it is widely used in nonlinear complex optimization problems.

The particle swarm algorithm realizes the search function by continuously iterating and updating the position P and velocity V of each particle in the particle swarm. In a D-dimensional search space, the position of the $i$−th particle at time t is a D-dimensional vector, $P_i(t) = (p_{i1,t}, p_{i2,t}, \ldots, p_{iD,t})$. Similarly, the velocity of the $i$−th particle at time t is also one D-dimensional vector, $V_i(t) = (v_{i1,t}, v_{i2,t}, \ldots, v_{iD,t})$. In order to find the optimal solution, each particle moves to its historical optimal position pbest($i,t$) and the group optimal position gbest($t$) at their respective speeds. The calculation formula of pbest($i,t$) and gbest($t$) is as follows:

$$\text{pbest}(i, t) = \underset{k=1,\ldots,t}{\arg\min}[f(P_i(k))], \quad i \in \{1, 2, \ldots, N_P\}; \tag{12}$$

$$\text{gbest}(t) = \underset{\substack{i=1,\ldots,N_P \\ k=1,\ldots,t}}{\arg\min}[f(P_i(k))]. \tag{13}$$

$i$ is the particle number, $N_P$ is the total number of particles in the particle swarm, t is the current iteration number, $f(\cdot)$ is the fitness function, and $P$ is the particle's position. The following equations update the position $P$ and velocity $V$ of each particle:

$$V_i(t + 1) = \omega V_i(t) + c_1 r_1(pbest(i, t) - P_i(t)) + c_2 r_2(gbest(t) - P_i(t)), \tag{14}$$

$$P_i(t+1) = P_i(t) + V_i(t+1). \tag{15}$$

$V$ is the particle update speed, $\omega$ is the inertial weight that weighs the algorithm's global and local optimization capabilities, $r_1$ and $r_2$ are random variables subjected to a uniform distribution in the interval [0,1], and $c_1$ and $c_2$ are acceleration coefficients. Normally, we add an upper bound to $V$ to prevent particles from leaving the search space. The particle velocity update Eq (12) can be divided into three parts. The first part is the inertia part, and the particles retain a part of the previous period's velocity inertia to roam in the entire search space. The second part is the "self-recognition" part, particles can continue to approach the optimal position of their historical iteration. The third part is the "group cooperation" part so that particles can continue to approach the particle group's optimal position. Input each particle's position into the fitness function $f(\cdot)$ to obtain the corresponding fitness value. The particle swarm algorithm aims to minimize the fitness function $f(\cdot)$ and obtain the global minimum fitness value and the corresponding particle position.

## 2.5. The PSO-MIDAS model

Given the significance of the selection of the lag structures in improving the forecast accuracy of the MIDAS model, in this section, we apply the PSO algorithm to the optimization of the lag structures of the MIDAS model. Specifically, we set the lag structures of the MIDAS model as the particle's position $P_i(t)$, the dimension of the particle's position is determined by the number of variables of the MIDAS model, and the lag structure of the MIDAS model with N variables is 1+N dimension, that is, the sum of the autoregressive term of the dependent variable and the number of the independent variables. Meanwhile, we set the fitness function $f(\cdot)$ as the RMSE of the MIDAS model on the validation set. Taking the univariate $ADL - MIDAS(p_Y^Q, q_X^M, L_X^M)$ model as an example, first divide the samples into the training set, validation set, and test set, input the two-dimensional lag structure $(p_Y^Q, q_X^M)$, estimate the parameters of the $ADL - MIDAS(p_Y^Q, q_X^M, L_X^M)$ model on the training set, and then obtain the forecast metrics of the model on the validation set. Then substituting the above process into the particle swarm optimization algorithm, we can find the MIDAS model with the best forecast metric on the validation set and expect that the model also has excellent prediction ability on the test set.

It is worth noting that we set the fitness function $f(\cdot)$ as the forecast metrics of the MIDAS model on the validation set rather than the training set because time series prediction is time-sensitive. The validation set, closer to the test set, is more timely than the training set so that the optimal lag structure can be extracted more effectively. Facing the complex nonlinear optimization problem of the MIDAS model's lag structures studied in this paper, we have the option of using various heuristic algorithms besides the PSO algorithm, such as the well-known genetic algorithm and simulated annealing algorithm. However, when compared to the PSO algorithm, both of these alternatives consume more computing resources. Additionally, the genetic algorithm is better suited for solving discrete optimization problems rather than the continuous optimization problems presented in this paper. In general, the introduction of the PSO algorithm can save the researcher's calculation and time costs while ensuring the global optimum solution. The pseudo-code of the PSO-MIDAS model can be summarised as follows:

```
1: for i in 1 to N_P do {for each particle in the swarm}
2: Randomly initialize particles' positions P_i(0) and velocities V_i(0)
3: Initialize particles' personal best pbest (i,0) and group best
gbest (0)
```

```
4: end for
5: repeat
6: for i in 1 to N_P do
7: Update particle's velocity using Eq (12)
8: Update particle's position using Eq (13)
9: if P_i(t)<pbest (i,t) then {minimization of f(·)}
10: Update particle's best-known position pbest (i,t+1) = P_i(t)
11: if pbest (i,t)<gbest (t) then {minimization of f(·)}
12: Update the group's best-known position gbest (t+1) = pbest (i,t)
13: end if
14: end if
15: end for
16: until [number of iterations T is met]
17: return gbest (T) {the optimal lag structure of the MIDAS model}
```

## 3. Empirical experiment

### 3.1. Data collection

We collect U.S. quarterly real GDP data as low-frequency data, and the sample interval is from the first quarter of 1968 to the first quarter of 2020. Based on existing economic and financial theory and literature [21], we select the three variables of U.S. Industrial Production (IP), Non-Farm Payroll (NFP), and Capacity Utilization (CU) as the predictor. The first two predictors are the components of the Conference Board Coincident Index. The three predictors are all monthly data used as high-frequency data in this article, and the sample interval is from January 1968 to March 2020. The U.S. GDP data comes from the Bureau of Economic Analysis. The GDP data and capacity utilization data come from the Federal Reserve Board, and the Non-Farm Payroll data comes from the Bureau of Labor Statistics. The descriptive statistics of the variables are shown in Table 1:

Fig 1 shows that the fluctuations in the quarterly GDP growth rate have significant periodicity. The GDP growth rate is positive in most cases, which means that the volatility of GDP rises. During the depression phase of the economic cycle, the GDP growth rate fell to a negative value. The most recent trough was around the 2008 global financial crisis, and there was no sign of gradual recovery until the end of 2009.

Fig 2 shows that the fluctuation of the monthly growth rate of industrial production (IP) is also cyclical, and the fluctuation range is relatively broad. The bottom value of several fluctuations is around -5%. After the 2008 financial crisis, the negative growth rate was even close to -15%. Compared with industrial production data, Non-Farm Payroll (NFP) monthly growth rate fluctuations are more cyclical and consistent with the changing GDP appreciation rate, reflecting the close relationship between employment and output. This indicator's fluctuation range is relatively small and is smaller than the fluctuation range of GDP appreciation rate. The monthly growth rate of capacity utilization (CU) fluctuates cyclically and fluctuates widely. The historical fluctuations in the sample interval are mainly concentrated around -15% to 10%.

**Table 1. Descriptive statistics of the variables.**

| Variables | Frequency | Min | Median | Mean | Max | Sd | Skewness | Kurtosis |
|---|---|---|---|---|---|---|---|---|
| GDP | quarterly | -3.979 | 2.971 | 2.829 | 8.578 | 2.123 | -0.470 | 0.860 |
| IP | monthly | -15.193 | 2.578 | 2.183 | 11.834 | 4.552 | -0.951 | 1.815 |
| NFP | monthly | -4.856 | 1.798 | 1.620 | 5.409 | 1.853 | -0.793 | 1.057 |
| CU | monthly | -16.253 | 0.326 | -0.091 | 10.909 | 4.439 | -0.739 | 1.638 |

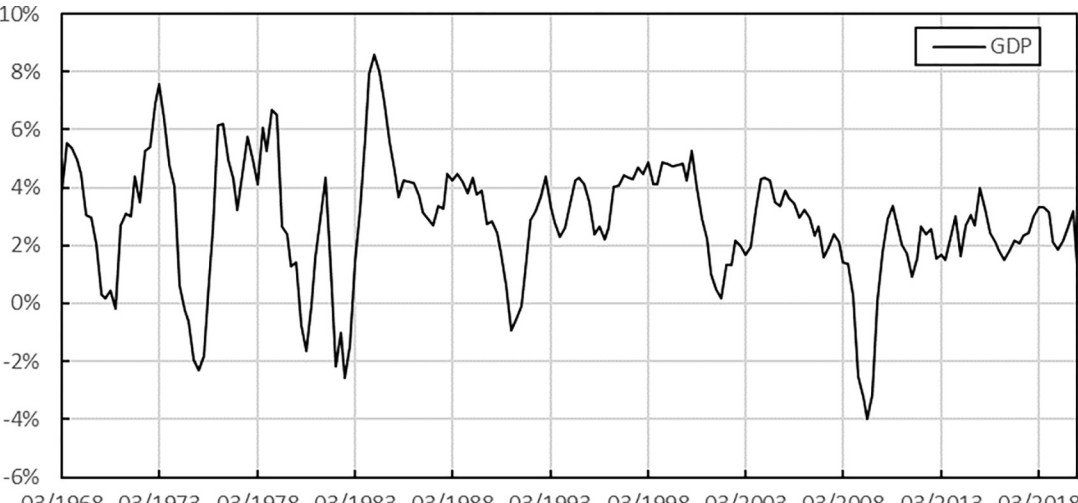

**Fig 1. Quarterly GDP growth rate.**

In summary, the three forecast factors are procyclical with GDP growth rate and change in the same direction. The fluctuation range of NFP is relatively small. Moreover, CU fluctuates wildly.

## 3.2. Empirical design

The empirical design includes two parts. The first part explores the empirical results of the univariate PSO-MIDAS model. Comparing the empirical results of the benchmark models verifies whether the optimization effect of the PSO algorithm on the univariate MIDAS model is significant. The second part further explores the empirical results of the multivariate PSO-MV--MIDAS model. Similarly, by comparing with the benchmark models' empirical results, verify whether the PSO algorithm's optimization effect on the multivariate MV-MIDAS model is

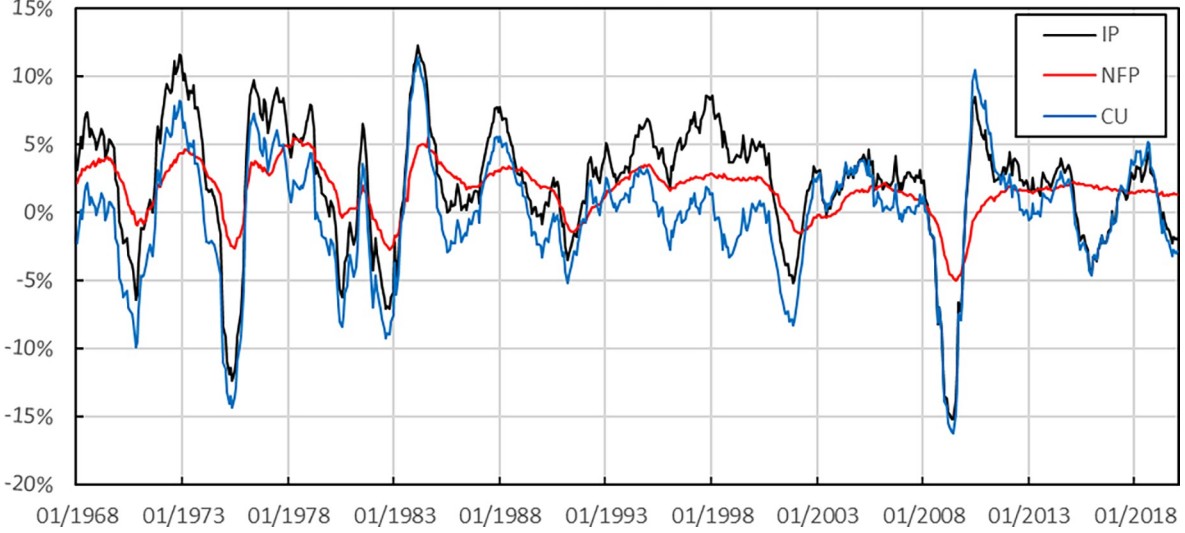

**Fig 2. Monthly growth rate of each factor.**

significant. Meanwhile, we explored whether there is any difference in the PSO algorithm's optimization effect on the univariate and multivariate MIDAS models. Before presenting the empirical results, we will give specific explanations on the related setting details of the benchmark models, MIDAS model, and PSO algorithm.

The benchmark models include ADL, U-MIDAS and ADL-MIDAS models (for convenience, the ADL-MIDAS model will be referred to as the MIDAS model below). The lag structures of the benchmark model are determined by the AIC information criterion method. Both the PSO-MIDAS model and the PSO-MV-MIDAS model use the PSO algorithm to determine the MIDAS model's optimal lag structure. The univariate model included the three monthly variables (IP, NFP, and CU) as the only high-frequency explanatory variable. The multivariate model included all the three high-frequency monthly variables as the high-frequency explanatory variables.

The forecasts of all models in this paper are out-of-sample rolling forecast, where the ratio of the training set to the validation set and test set in the sample is 7:1:2. We use the PSO algorithm to find the optimal lag structure based on the model's RMSE on the validation set. The model's RMSFE and MAPE on the test set is the average of the 42 rolling forecast metrics. The value range of the forecast horizon $h$ is four quarters, that is, $h = 1,2,3,4$. Each quarter contains three monthly leads, namely $L_X^M = 0, 1, 2$, we total study 12 scenarios: use current quarter's data for real-time nowcasting and use one to three quarters' data in advance for forwarding forecasting, with a total period of 12 months. It is worth noting that, since the ADL model can not include the lead information of high-frequency monthly data, we only examined the ADL model under four forecast horizons without leads.

Fig 3 shows the empirical design process of the PSO-MIDAS flowchart, the algorithm programming is carried out via The MIDAS toolbox for R (midasr package) [26]. Regarding the PSO algorithm's specific hyperparameter setting, we refer to the standard PSO algorithm's settings of related hyperparameters in the Standard Particle Swarm Optimisation 2011 (SPSO 2011) proposed by Clerc et al. [27]. Readers can find more details about the PSO hyperparameter settings in the Standard Particle Swarm Optimisation [28]. The optimization interval of the PSO algorithm is set to 5 years (20 quarters, 60 months), that is, the low-frequency lags of GDP $p_{GDP}^Q \in [1, 20]$, the high-frequency lags of predictors $p_X^M \in [1, 60]$. The PSO algorithm hyperparameter settings in this article are shown in Table 2:

## 4. Results

### 4.1. Univariate PSO-MIDAS model

Univariate models refer to models that use only one predictor variable as the explanatory variable. Therefore, we use IP, NFP, and CU indicators to construct three types of PSO-MIDAS models and benchmark models. In Table 3, the first three columns describe the matching relationship with different forecast horizon h, lead, and forecast type. Using the current quarter's monthly data to forecast the current quarter's variable is called real-time forecasting. Using the previous quarter's monthly data to forecast is the forward forecast in the usual sense. The result of the PSO-MIDAS model is the RMSFE value of out-of-sample rolling forecasts, while the MIDAS model, U-MIDAS model and ADL model are the benchmark models. We compared the PSO-MIDAS model's RMSFE value with the benchmark models' RMSFE value and used the ratio as the model comparison result. If the ratio is less than 100%, the RMSFE of the PSO-MIDAS model is smaller than the RMSFE of the benchmark model, and the forecast capability of the PSO-MIDAS model is better than benchmark models. Conversely, if the ratio is greater than 100%, it indicates that the PSO-MIDAS model's forecast capability has not improved. The smaller the ratio, the better the PSO-MIDAS optimization effect. In the last row of the

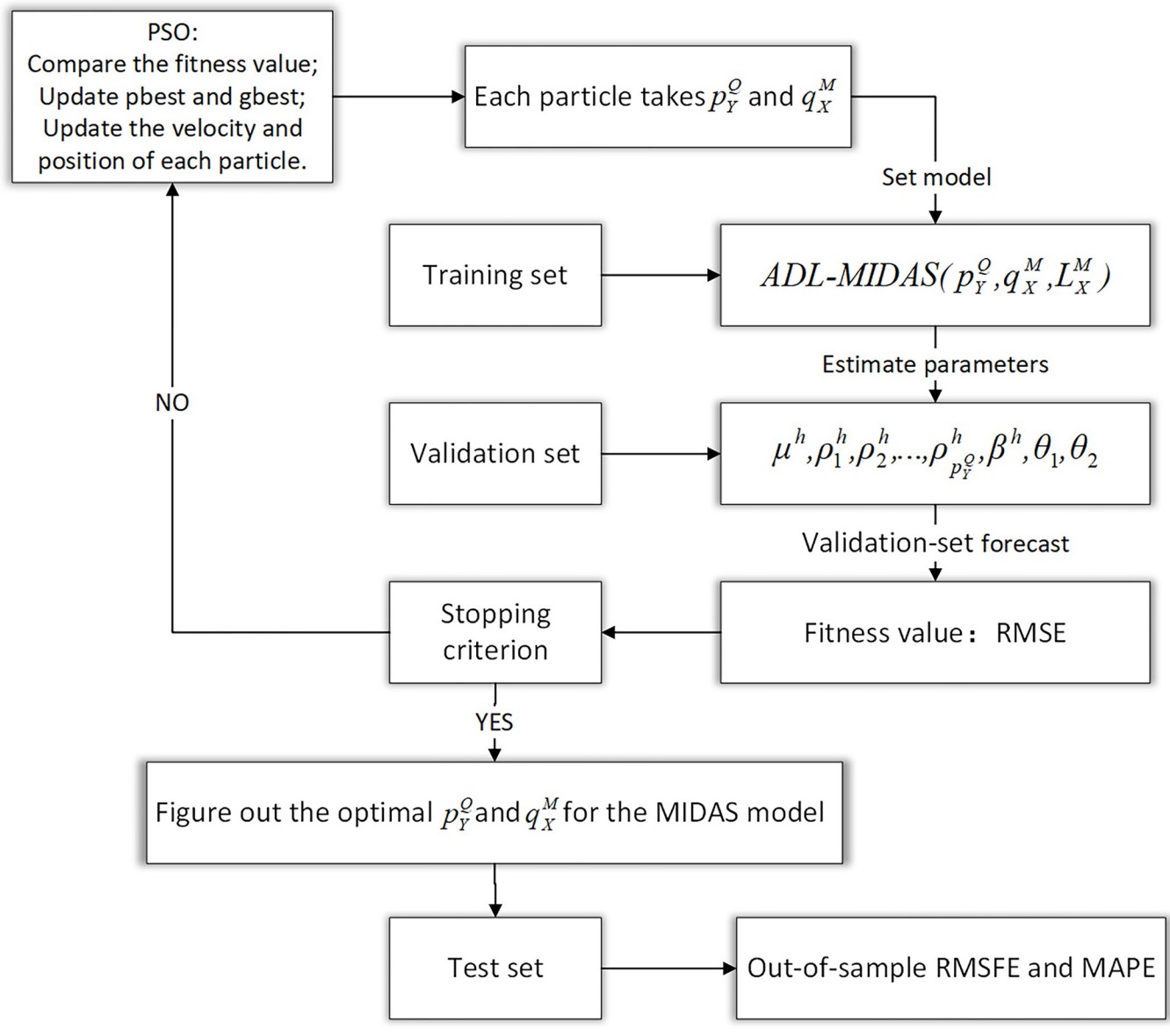

**Fig 3. The flowchart of the PSO-MIDAS model.**

**Table 2. Hyperparameters' setting for PSO algorithm.**

| Hyperparameter | value |
|---|---|
| Swarm size ($N_P$) | 40 |
| Max iterations | 100 |
| Inertia weight ($\omega$) | 1/(2*ln(2)) |
| Acceleration coefficient ($c_1$) | 0.5+ln(2) |
| Acceleration coefficient ($c_2$) | 0.5+ln(2) |
| Particles' position bound ($p_{GDP}^Q$) | [1,20] |
| Particles' position bound ($p_{IP}^M, p_{NFP}^M, p_{CU}^M$) | [1,60] |

**Table 3. Comparison of the RMSFE of univariate mixed-frequency models: Based on IP indicators.**

| Horizon | h | Lead | PSO-MIDAS (RMSFE) | Ratio to | | |
|---|---|---|---|---|---|---|
| | | | | MIDAS | U-MIDAS | ADL |
| Nowcast | 1 | 2 | 0.6832 | 98.88% | 82.85% | 94.88% |
| | | 1 | 0.6551 | 96.33% | 78.87% | 90.97% |
| | | 0 | 0.6431 | 96.45% | 77.04% | 57.59% |
| Forecast | 2 | 2 | 1.0956 | 95.89% | 87.81% | 98.10% |
| | | 1 | 1.0268 | 93.06% | 74.22% | 91.95% |
| | | 0 | 0.8982 | 86.36% | 57.42% | 63.18% |
| | 3 | 2 | 1.1987 | 80.28% | 61.98% | 84.32% |
| | | 1 | 1.1186 | 77.58% | 54.12% | 78.68% |
| | | 0 | 1.0234 | 77.95% | 44.62% | 52.01% |
| | 4 | 2 | 1.2982 | 71.42% | 49.43% | 65.98% |
| | | 1 | 1.3139 | 73.63% | 48.64% | 66.78% |
| | | 0 | 1.2959 | 72.83% | 44.21% | 59.53% |
| Average | | | | 85.05% | 63.43% | 75.33% |

table, we calculated the average ratio of the RMSFE for the PSO-MIDAS model compared to three benchmark models to evaluate the average optimization performance of the PSO-MI-DAS model.

Table 3 shows that when IP as the explanatory variable, the PSO-MIDAS model's forecast capability is better than other benchmark models. First of all, the PSO-MIDAS model is significantly better than the U-MIDAS model. Compared with the U-MIDAS model's RMSFE level, the optimization effect of PSO-MIDAS is 37% on average. Secondly, the PSO-MIDAS model's forecast capability is also better than the ADL models. The optimization effect is about 25%, indicating that the use of high-frequency data of the explanatory variable IP has contributed to the GDP forecast. Finally, the PSO-MIDAS model's forecast results are better than the MIDAS model, the RMSFE ratio is less than 1, and the improvement effect is about 15%. It is worth noting that when h becomes larger, the RMSFE of the model also becomes larger, suggesting that using the same information to predict the more distant future will be more difficult than predicting the more recent future. However, when h is large, the PSO-MIDAS is better optimised for forecast capability relative to the benchmark models. This suggests that when h is large, the predictive performance of the MIDAS model becomes more sensitive to the choice of lag structure, while the PSO-MIDAS model constructed in this paper solves the problem of choosing the MIDAS model's mixed-frequency lag structure. The particle swarm algorithm improves the traditional MIDAS model, which guarantees adequate and sufficient high-frequency information and filters the redundant data noise.

In Table 4, we replace the forecast metric in Table 3 with the MAPE (Mean Absolute Percentage Error) to evaluate the forecast accuracy of the model on the test set. The calculation formula of MAPE is as follow:

$$MAPE = \sqrt{E[|Y_{t+h} - \hat{Y}_{t+h|t}|/|Y_{t+h}|]} \tag{16}$$

The comparison results are similar to those in Table 3. The optimization effects of the PSO-MIDAS model relative to the MIDAS, U-MIDAS, and ADL models are 13%, 32%, and 23% respectively. We notice that samples with larger forecast errors have a greater impact on the MAPE compared to the RMSE. Therefore, the optimization effect of the PSO-MIDAS model, as measured by MAPE, is slightly inferior. Moreover, we assess the significance of

**Table 4. Comparison of the MAPE of univariate mixed-frequency models: Based on IP indicators.**

| Horizon | h | Lead | PSO-MIDAS (MAPE) (%) | Ratio to | | |
|---|---|---|---|---|---|---|
| | | | | MIDAS | U-MIDAS | ADL |
| Nowcast | 1 | 2 | 22.82 | **102.20%** | 85.93% | **106.92%** |
| | | 1 | 21.65 | 98.35% | 81.80% | **101.45%** |
| | | 0 | 22.40 | **102.32%** | 79.89% | 58.59% |
| Forecast | 2 | 2 | 38.17 | 95.34% | 98.00% | 99.86% |
| | | 1 | 35.83 | 98.04% | 84.62% | 93.73% |
| | | 0 | 31.00 | 89.86% | 60.57% | 63.11% |
| | 3 | 2 | 40.47 | 82.50% | 63.76% | 82.38% |
| | | 1 | 37.53 | 83.57% | 56.80% | 76.40% |
| | | 0 | 35.09 | 80.85% | 46.86% | 53.20% |
| | 4 | 2 | 42.56 | 74.88% | 56.36% | 64.52% |
| | | 1 | 40.64 | 71.91% | 52.11% | 61.61% |
| | | 0 | 39.06 | 69.78% | 46.49% | 56.36% |
| Average | | | | 87.47% | 67.77% | 76.51% |

differences in forecast accuracy using the Diebold and Mariano test and Harvey et al.'s small-sample bias-corrected variance [29, 30]. The Diebold-Mariano Test compares the accuracy of two forecasting methods by assessing whether the difference in forecast errors is statistically significant. The results are reported in Table 5. The Diebold–Mariano test results show that the forecast accuracy of the PSO-MIDAS model is significantly better than that of the U-MIDAS model, and when h = 3, 4, the forecast accuracy of the PSO-MIDAS model is significantly better than that of the MIDAS model and the ADL model.

Table 6 shows that when NFP is the explanatory variable, the PSO-MIDAS model's RMSFE is better than other benchmark models. Compared with the U-MIDAS model, the average optimization effect of PSO-MIDAS is about 9%. The PSO-MIDAS model's forecast capability is also better than the ADL and MIDAS model, and the optimization effect is about 23% and 15%, respectively. Although the PSO-MIDAS model's forecast capability using NFP as the explanatory variable is generally better than the benchmark model, its optimization effect is

**Table 5. Diebold–Mariano test results: Based on IP indicators.**

| h | Lead | PSO-MIDAS vs. MIDAS | PSO-MIDAS vs. U-MIDAS | PSO-MIDAS vs. ADL |
|---|---|---|---|---|
| 1 | 2 | 0.43 | **0.04** | 0.79 |
| | 1 | 0.33 | **0.03** | 0.66 |
| | 0 | 0.31 | **0.01** | 0.63 |
| 2 | 2 | 0.24 | 0.14 | 0.44 |
| | 1 | 0.23 | **0.04** | 0.29 |
| | 0 | 0.13 | **0.00** | **0.07** |
| 3 | 2 | **0.01** | **0.00** | 0.12 |
| | 1 | **0.03** | **0.00** | **0.04** |
| | 0 | **0.03** | **0.00** | **0.01** |
| 4 | 2 | **0.00** | **0.00** | **0.02** |
| | 1 | **0.00** | **0.00** | **0.00** |
| | 0 | **0.00** | **0.00** | **0.00** |

Notes: Bold values indicate where the null hypothesis can be rejected at a 10% significance. Results report p-value for H0 = model A have lower accuracy than model B over 2009Q4–2020Q1.

**Table 6. Comparison of the RMSFE of univariate mixed-frequency models: Based on NFP indicators.**

| Horizon | h | Lead | PSO-MIDAS (RMSFE) | Ratio to | | |
|---|---|---|---|---|---|---|
| | | | | MIDAS | U-MIDAS | ADL |
| Nowcast | 1 | 2 | 0.6149 | 94.02% | 93.67% | 84.26% |
| | | 1 | 0.6347 | 97.86% | 98.69% | 86.98% |
| | | 0 | 0.6281 | 97.64% | 97.46% | 64.22% |
| Forecast | 2 | 2 | 0.8891 | 85.28% | **104.37%** | 90.91% |
| | | 1 | 0.8689 | 84.61% | 99.34% | 88.84% |
| | | 0 | 0.8585 | 86.40% | 98.07% | 66.19% |
| | 3 | 2 | 1.0412 | 77.89% | 81.00% | 80.28% |
| | | 1 | 1.0228 | 77.17% | 76.80% | 78.87% |
| | | 0 | 1.0924 | 84.48% | 90.13% | 60.91% |
| | 4 | 2 | 1.2276 | 71.00% | 76.61% | 68.45% |
| | | 1 | 1.3779 | 78.24% | 82.03% | 76.82% |
| | | 0 | 1.4241 | 80.54% | 89.32% | 73.36% |
| Average | | | | 84.59% | 90.62% | 76.67% |

inferior to the PSO-MIDAS model using IP as the explanatory variable. Tables 7 and 8 report the MAPE and Diebold–Mariano test results of the PSO-MIDAS and the benchmark model, respectively. The Diebold–Mariano test results show that the optimization effect of the PSO-MIDAS is not significant compared to the U-MIDAS, and when h = 3, 4, the forecast accuracy of the PSO-MIDAS model is significantly better than that of the MIDAS model and the ADL model.

Furthermore, Table 9 shows that when CU as the explanatory variable, the RMSFE results of the PSO-MIDAS model is still better than other benchmark models. Compared with the U-MIDAS model, the average optimization effect is 25%; compared with the ADL model, the average optimization effect is 19%; compared with the MIDAS model, the average optimization effect is 12%. The MAPE and Diebold–Mariano test results in Tables 10 and 11 are similar to those of the PSO-MIDAS model with IP variable. In conclusion, the PSO-MIDAS model's forecast accuracy is better than the benchmark univariate model, especially when the forecast

**Table 7. Comparison of the MAPE of univariate mixed-frequency models: Based on NFP indicators.**

| Horizon | h | Lead | PSO-MIDAS (MAPE) (%) | Ratio to | | |
|---|---|---|---|---|---|---|
| | | | | MIDAS | U-MIDAS | ADL |
| Nowcast | 1 | 2 | 20.90 | 97.24% | **113.44%** | **102.77%** |
| | | 1 | 20.28 | 96.59% | **114.57%** | 99.72% |
| | | 0 | 20.89 | **103.51%** | **112.88%** | 64.59% |
| Forecast | 2 | 2 | 34.87 | 94.32% | **120.76%** | **107.79%** |
| | | 1 | 33.77 | 96.69% | **122.26%** | **104.38%** |
| | | 0 | 30.19 | 89.83% | **106.05%** | 72.23% |
| | 3 | 2 | 36.09 | 76.03% | 84.88% | 86.36% |
| | | 1 | 32.97 | 72.73% | 79.00% | 78.89% |
| | | 0 | 37.61 | 87.18% | 97.25% | 68.97% |
| | 4 | 2 | 40.83 | 69.95% | 78.14% | 74.88% |
| | | 1 | 41.44 | 73.17% | 79.05% | 76.00% |
| | | 0 | 43.23 | 78.42% | 92.57% | 73.20% |
| Average | | | | 86.31% | 100.07% | 84.15% |

**Table 8. Diebold–Mariano test results: Based on NFP indicators.**

| h | Lead | PSO-MIDAS vs. MIDAS | PSO-MIDAS vs. U-MIDAS | PSO-MIDAS vs. ADL |
|---|------|---------------------|----------------------|-------------------|
| 1 | 2 | 0.20 | 0.79 | 0.41 |
|   | 1 | 0.39 | 0.85 | 0.53 |
|   | 0 | 0.39 | 0.87 | 0.49 |
| 2 | 2 | 0.37 | 0.89 | 0.59 |
|   | 1 | 0.40 | 0.81 | 0.56 |
|   | 0 | 0.12 | 0.44 | 0.15 |
| 3 | 2 | **0.02** | 0.11 | **0.07** |
|   | 1 | **0.02** | **0.10** | **0.04** |
|   | 0 | **0.09** | 0.27 | **0.08** |
| 4 | 2 | **0.00** | **0.06** | **0.01** |
|   | 1 | **0.01** | 0.18 | **0.00** |
|   | 0 | **0.01** | 0.19 | **0.00** |

horizon *h* is large. After we embedded the particle swarm algorithm in the MIDAS model, we can efficiently determine the MIDAS model's lag structure and improve the prediction accuracy of the model.

## 4.2. Multivariate PSO-MV-MIDAS model

Whether it is a univariate PSO-MIDAS model or a multivariate PSO-MV-MIDAS model, it essentially optimizes the selection of mixed-frequency lag structure in the MIDAS model through the PSO algorithm. However, the number of variables in the MIDAS model changes the optimization task of the PSO algorithm. The univariate PSO-MIDAS model solves a two-dimensional optimization problem. The multivariate PSO-MV-MIDAS model incorporates three variables: IP, NFP, and CU. Including the auto-regressive term of quarterly GDP growth rate, the PSO-MV-MIDAS model solves a four-dimensional optimization problem.

Tables 12 and 13 show that the PSO-MV-MIDAS model outperforms other multivariate benchmark models in terms of forecast accuracy, as measured by RMSFE and MAPE.

**Table 9. Comparison of the RMSFE of univariate mixed-frequency models: Based on CU indicators.**

| Horizon | h | Lead | PSO-MIDAS (RMSFE) | Ratio to | | |
|---------|---|------|-------------------|----------|---------|---------|
|         |   |      |                   | MIDAS | U-MIDAS | ADL |
| Nowcast | 1 | 2 | 0.6523 | 89.92% | 79.26% | 91.16% |
|         |   | 1 | 0.6230 | 88.66% | 76.62% | 87.08% |
|         |   | 0 | 0.6491 | 95.73% | 79.83% | 63.00% |
| Forecast | 2 | 2 | 0.9463 | **103.30%** | 77.68% | 91.85% |
|          |   | 1 | 1.0060 | 93.67% | 85.66% | 97.64% |
|          |   | 0 | 0.9323 | 94.79% | 73.88% | 73.61% |
|          | 3 | 2 | 1.0971 | 92.34% | 74.68% | 86.62% |
|          |   | 1 | 1.1028 | 83.64% | 74.09% | 87.07% |
|          |   | 0 | 0.9764 | 80.02% | 61.68% | 57.35% |
|          | 4 | 2 | 1.4142 | 77.65% | 76.91% | 83.06% |
|          |   | 1 | 1.4148 | 76.27% | 71.25% | 83.09% |
|          |   | 0 | 1.4036 | 82.21% | 66.79% | 76.25% |
| Average |   |   |   | 88.18% | 74.86% | 81.48% |

**Table 10. Comparison of the MAPE of univariate mixed-frequency models: Based on CU indicators.**

| Horizon | h | Lead | PSO-MIDAS (MAPE) (%) | Ratio to | | |
|---|---|---|---|---|---|---|
| | | | | MIDAS | U-MIDAS | ADL |
| Nowcast | 1 | 2 | 20.98 | 92.63% | 79.06% | 97.30% |
| | | 1 | 19.99 | 89.67% | 77.39% | 92.72% |
| | | 0 | 21.17 | 96.61% | 80.35% | 59.28% |
| Forecast | 2 | 2 | 37.13 | 97.58% | **101.01%** | **103.96%** |
| | | 1 | 30.96 | 87.91% | 85.10% | 86.69% |
| | | 0 | 31.63 | **100.30%** | 78.35% | 72.32% |
| | 3 | 2 | 36.14 | 77.11% | 77.64% | 82.64% |
| | | 1 | 36.59 | 87.66% | 76.01% | 83.66% |
| | | 0 | 34.04 | 80.66% | 66.01% | 61.19% |
| | 4 | 2 | 44.71 | 80.92% | 75.33% | 80.37% |
| | | 1 | 42.66 | 75.68% | 70.59% | 76.68% |
| | | 0 | 41.82 | 74.19% | 67.39% | 70.92% |
| Average | | | | 86.74% | 77.85% | 80.64% |

**Table 11. Diebold–Mariano test results: Based on CU indicators.**

| h | Lead | PSO-MIDAS vs. MIDAS | PSO-MIDAS vs. U-MIDAS | PSO-MIDAS vs. ADL |
|---|---|---|---|---|
| 1 | 2 | **0.04** | **0.05** | 0.66 |
| | 1 | **0.03** | **0.03** | 0.53 |
| | 0 | 0.21 | **0.06** | 0.66 |
| 2 | 2 | 0.36 | 0.37 | 0.74 |
| | 1 | 0.18 | 0.16 | 0.44 |
| | 0 | 0.56 | **0.01** | 0.20 |
| 3 | 2 | **0.01** | 0.12 | 0.36 |
| | 1 | 0.11 | **0.05** | 0.17 |
| | 0 | **0.01** | **0.00** | **0.02** |
| 4 | 2 | **0.02** | **0.03** | **0.09** |
| | 1 | **0.00** | **0.01** | **0.01** |
| | 0 | **0.00** | **0.01** | **0.02** |

**Table 12. Comparison of the RMSFE of multivariate mixed-frequency models: Based on IP, NFP, and CU indicators.**

| Horizon | h | Lead | PSO-MV-MIDAS (RMSFE) | Ratio to | | |
|---|---|---|---|---|---|---|
| | | | | MV-MIDAS | U-MV-MIDAS | MV-ADL |
| Nowcast | 1 | 2 | 0.6794 | 99.19% | 35.04% | 78.03% |
| | | 1 | 0.7217 | **105.62%** | 35.49% | 82.90% |
| | | 0 | 0.6266 | 94.74% | 36.56% | 51.36% |
| Forecast | 2 | 2 | 0.9593 | 87.36% | 33.26% | 78.63% |
| | | 1 | 0.9842 | 90.05% | 26.33% | 80.67% |
| | | 0 | 0.9353 | 88.94% | 30.57% | 56.30% |
| | 3 | 2 | 1.1853 | 81.01% | 23.83% | 71.34% |
| | | 1 | 1.1124 | 77.92% | 23.14% | 66.95% |
| | | 0 | 1.2549 | 92.83% | 28.31% | 51.18% |
| | 4 | 2 | 1.3272 | 74.47% | 23.52% | 54.13% |
| | | 1 | 1.5382 | 83.50% | 26.04% | 62.73% |
| | | 0 | 1.5261 | 82.02% | 30.32% | 53.37% |
| Average | | | | 88.14% | 29.37% | 65.63% |

**Table 13. Comparison of the MAPE of multivariate mixed-frequency models: Based on IP, NFP, and CU indicators.**

| Horizon | h | Lead | PSO-MV-MIDAS (MAPE) (%) | Ratio to | | |
|---|---|---|---|---|---|---|
| | | | | MV-MIDAS | U-MV-MIDAS | MV-ADL |
| Nowcast | 1 | 2 | 20.35 | 97.12% | 33.90% | 81.39% |
| | | 1 | 23.84 | **110.77%** | 36.85% | 95.36% |
| | | 0 | 20.95 | 96.42% | 39.39% | 49.03% |
| Forecast | 2 | 2 | 33.78 | 87.47% | 36.19% | 79.04% |
| | | 1 | 35.40 | 96.23% | 29.68% | 82.83% |
| | | 0 | 31.28 | 89.52% | 29.80% | 52.67% |
| | 3 | 2 | 39.79 | 82.91% | 27.10% | 67.01% |
| | | 1 | 35.37 | 78.62% | 23.37% | 59.56% |
| | | 0 | 42.28 | 97.09% | 30.27% | 51.61% |
| | 4 | 2 | 40.83 | 73.86% | 24.58% | 49.84% |
| | | 1 | 46.77 | 85.76% | 27.50% | 57.10% |
| | | 0 | 44.16 | 77.64% | 27.73% | 45.33% |
| Average | | | | 89.45% | 30.53% | 64.23% |

Compared with the U-MV-MIDAS model, the average optimization effect is 71%; compared with the MV-ADL model, the average optimization effect is 34%; compared with the MV-MIDAS model, the average optimization effect is 12%. Similar with univariate PSO-MIDAS models, the PSO-MV-MIDAS model has a better optimization effect when the forecast horizon h is larger. The Diebold–Mariano test results of the PSO-MV-MIDAS and the multivariate benchmark model in Table 14 show that the forecast accuracy of the PSO- MV-MIDAS model is significantly better than that of the multivariate benchmark models when h = 2, 3, 4.

Finally, for better analysis of the result, we summarize the average optimization effects of the univariate and multivariate PSO-MIDAS models relative to the benchmark model into a histogram plot, as shown in Fig 4. When calculating the optimization effect of the PSO-MIDAS model, we exclude model results that are not significant in the Diebold–Mariano test. Regardless of the univariate and multivariate PSO-MIDAS models, their optimization effect relative to the MIDAS model is maintained at an average level of 10%. Additionally, we present the GDP prediction results of the PSO-MIDAS model with a forecast horizon of 7 months (the

**Table 14. Diebold–Mariano test results: Based on IP, NFP, and CU indicators.**

| h | Lead | PSO- MV-MIDAS vs. MV-MIDAS | PSO- MV-MIDAS vs. U- MV-MIDAS | PSO- MV-MIDAS vs. MV-ADL |
|---|---|---|---|---|
| 1 | 2 | 0.79 | **0.00** | 0.21 |
| | 1 | 0.81 | **0.00** | 0.33 |
| | 0 | 0.25 | **0.00** | **0.08** |
| 2 | 2 | **0.07** | **0.00** | **0.01** |
| | 1 | 0.19 | **0.00** | **0.03** |
| | 0 | **0.08** | **0.00** | **0.01** |
| 3 | 2 | **0.02** | **0.00** | **0.01** |
| | 1 | **0.07** | **0.00** | **0.00** |
| | 0 | 0.20 | **0.00** | **0.02** |
| 4 | 2 | **0.00** | **0.00** | **0.00** |
| | 1 | **0.04** | **0.00** | **0.01** |
| | 0 | **0.01** | **0.00** | **0.01** |

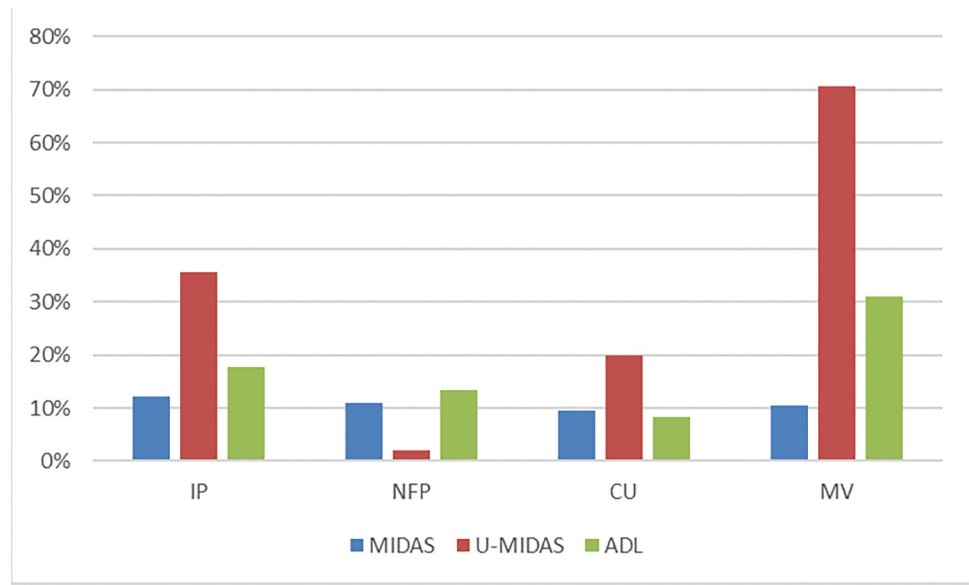

**Fig 4. The average optimization effects of the PSO-MIDAS against three benchmark models.**

Diebold–Mariano test results show that the univariate and multivariate PSO-MIDAS models outperform the benchmark MIDAS model when forecast horizon h = 7 months), as shown in Fig 5. It is worth noting that in multivariate models, the total lag order of the model grows exponentially, which means that the number of parameters to be estimated in the U-MV-MIDAS and MV-ADL model increases considerably, thus reducing the model's generalisation ability. This explains why the PSO-MV-MIDAS model significantly outperforms both the U-MV-MIDAS and MV-ADL models.

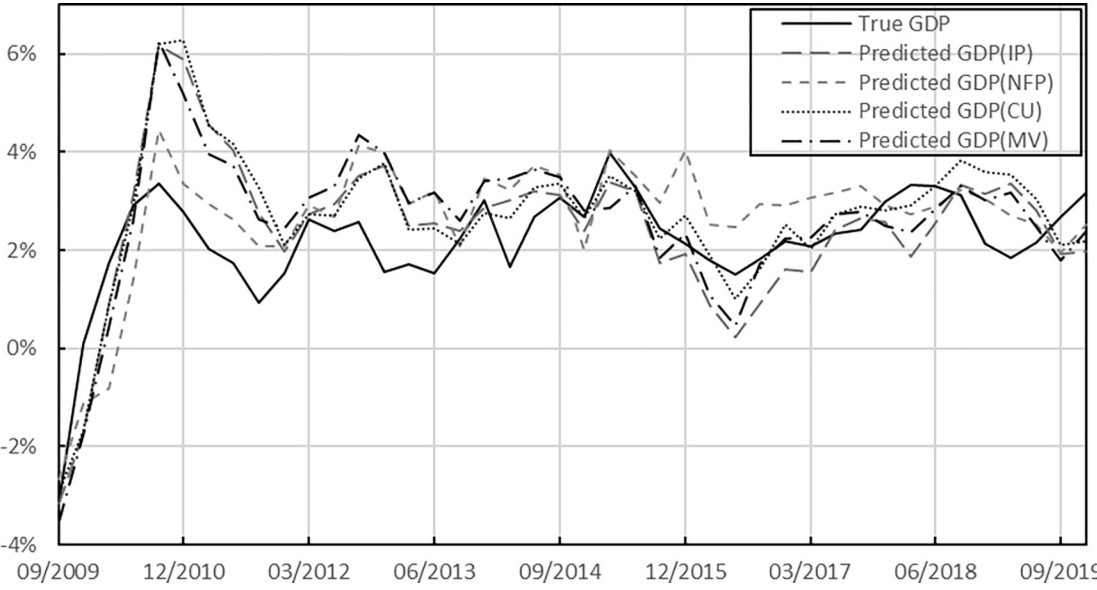

**Fig 5. Predicted GDP of the PSO-MIDAS model with a forecast horizon of 7 months.**

## 5. Conclusion

When utilizing the MIDAS model for GDP growth forecasting, the traditional model selection method cannot effectively determine the mixed-frequency lag structure, which impacts the accuracy of the model's forecasts. In this paper, the PSO-MIDAS model was proposed to resolve the MIDAS model's mixed-frequency lag structure selection problems. The PSO-MIDAS univariate model was shown to be better than the benchmark model MIDAS model, the U-MIDAS model and the ADL model, and the multivariate PSO-MV-MIDAS model was also found to perform better than the multivariate benchmark models. The Diebold–Mariano test results have shown that incorporating the PSO algorithm significantly enhances the forecasting ability of the MIDAS model, particularly for longer forecast horizons. Based on the research findings in this article, when applying the MIDAS model for macroeconomic forecasting, it is suggested to use the PSO-MIDAS model when the forecast horizon is greater than 2 quarters, and the PSO-MV-MIDAS model when the forecast horizon is greater than 1 quarter. In terms of selecting independent variables, the forecast performance of the PSO-MIDAS model with IP and CU as independent variables is superior to that of the PSO-MIDAS model with NFP as the independent variable. Additionally, the standard deviation of the NFP variable is significantly smaller than that of other variables, so it is recommended to use independent variables with larger standard deviations.

During the research, we also found some limitations of the PSO-MIDAS model, providing some directions for future research. First, the effectiveness of the PSO-MIDAS model is based on the assumption that the time series remains consistent in the validation set and the test set. If the potential lag structure of the target time series changes on the test set, we cannot use the validation set to calculate the potential optimal lag structure after the change. For example, our model had poor prediction results during the COVID-19 pandemic, which may be because the selected variables and their lag structures do not fully reflect the impact of the epidemic, which is also a problem faced by the benchmark MIDAS model. However, some studies have identified the inconsistency of time series through methods such as Regime Switch, which inspires the improvement of PSO-MIDAS. Second, we also tried to optimize the PSO-MIDAS lag structure through the validation set MAPE metric. Still, its model accuracy is not as good as the PSO-MIDAS model with the validation set RMSE as the fitness function. This may be because the MAPE metric is more sensitive to larger forecast errors, resulting in poor continuity of the fitness function, leading to poor PSO algorithm optimization. Therefore, the selection of forecast metrics for the fitness function is also a future research direction for PSO-MIDAS. Third, the PSO-MIDAS model can be implemented in various fields and data frequency mixtures in future studies. Including financial market variables could change the data frequency mixtures to monthly-daily or even quarterly-daily combinations. Exploring the optimization impact of the PSO-MIDAS model in different data frequency mixtures is an intriguing topic.

## Supporting information

**S1 File. Data and code used in this article.**
(ZIP)

## Author Contributions

**Conceptualization:** Feng Shen, Yuhuang Shang.

**Data curation:** Xiaodong Yan.

**Funding acquisition:** Feng Shen.

**Methodology:** Yuhuang Shang.

**Software:** Xiaodong Yan.

**Supervision:** Feng Shen, Yuhuang Shang.

**Validation:** Xiaodong Yan.

**Visualization:** Xiaodong Yan.

**Writing – original draft:** Feng Shen, Xiaodong Yan, Yuhuang Shang.

**Writing – review & editing:** Feng Shen, Xiaodong Yan, Yuhuang Shang.

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
