## [Decision Letter · Decision Letter 0]

1 Jul 2024

PONE-D-24-22830A novel hybrid PSO-MIDAS model and its application to the U.S. GDP forecastPLOS ONE

Dear Dr.  Shen,

Thank you for submitting your manuscript to PLOS ONE. After careful consideration, we feel that it has merit but does not fully meet PLOS ONE’s publication criteria as it currently stands. Therefore, we invite you to submit a revised version of the manuscript that addresses the points raised during the review process.

We look forward to receiving your revised manuscript.

Kind regards,

Sercan Ergün

Academic Editor

PLOS ONE

Journal Requirements:

   "This study was funded by the National Natural Science Foundation of China (no. 72001178), the project of Research Center for System Sciences and Enterprise Development (no. Xq22B03), and the Fundamental Research Funds for the Central Universities (no. JBK2304021)."

4. We note you have included a table to which you do not refer in the text of your manuscript. Please ensure that you refer to Table 1 in your text; if accepted, production will need this reference to link the reader to the Table.

Additional Editor Comments:

We have now received the reviewers’ reports, and after careful consideration, we have decided that your manuscript requires major revisions before it can be considered for publication.

Reviewers' comments:

Reviewer's Responses to Questions

**Comments to the Author**

1. Is the manuscript technically sound, and do the data support the conclusions?

Reviewer #1: Yes

Reviewer #2: Yes

Reviewer #3: Yes

2. Has the statistical analysis been performed appropriately and rigorously? 

Reviewer #1: Yes

Reviewer #2: Yes

Reviewer #3: No

3. Have the authors made all data underlying the findings in their manuscript fully available?

Reviewer #1: Yes

Reviewer #2: Yes

Reviewer #3: No

4. Is the manuscript presented in an intelligible fashion and written in standard English?

Reviewer #1: Yes

Reviewer #2: Yes

Reviewer #3: Yes

5. Review Comments to the Author

Reviewer #1: The following comments are provided for improving the manuscript:

1. The authors need to clearly explain what makes their work new and different, especially because there are already similar PSO-MIDAS models out there. Other researchers like Xiang (2018), Rosso and colleagues (2021) have worked on models of this kind.

[1] Xiang, C. (2018). Study on Forecast of Real Estate Prosperity Index Based on MIDAS Model Corrected by Mixed Kernel Function SVM. DOI: 10.12677/AAM.2018.712182

[2] Rosso, M. M., Cucuzza, R., Di Trapani, F., & Marano, G. C. (2021). Nonpenalty machine learning constraint handling using PSO‐SVM for structural optimization. Advances in Civil Engineering, 2021(1), 6617750.

2. The paper mainly compares its model with basic MIDAS models, but this doesn't show enough how the paper improves on what's already out there. So, the authors should also include how their model does compared to other well-known models, like those that predict things over time or models that look at data at the same frequency.

3. The paper needs to give a clearer and fuller explanation of how it combines the PSO algorithm with the MIDAS model, and what benefits this combination offers. The current explanation is too brief and doesn't show that the study's methods are strong and reliable.

4. The use of the PSO algorithm with only three explanatory variables should be justified. The authors need to clarify how the algorithm's benefits manifest in this context and why it was chosen over other variable selection methods.

5. The parameter settings listed in Table 1 need further elucidation. The authors should provide a rationale for these choices, supported by relevant references to enhance the credibility of their methodological approach.

6. The manuscript relies exclusively on RMSFE to evaluate forecasting accuracy, which is overly simplistic. Commonly used metrics such as MAPE and comparative tests like Diebold-Mariano tests are missing. Incorporating these would allow for a more nuanced assessment of the model's efficacy.

7. The conclusion should be expanded to include actionable policy recommendations and a discussion on the broader applications of the research. This would increase the paper's relevance and practical value.

Reviewer #2: The data used for the study are all provided and the statistical and/or data analysis carried out have all been done according to best practices with the requisite rigour. It is important, however, to consider the following observations:

1) More rigour in justifying the choice of PSO for the optimization specifically as against the general comments made in comparison with other possible techniques

2) How does the MIDAS approach in the work of Philip H. Franses (2016) "Yet Another Look at MIDAS Regression" relate (positively or otherwise) with that adopted in this work

3) There should be more rigour in describing and contrasting between the other Information Criterion approaches apart from the 2 mentioned (BIC & AIC) here, such as Hannan-Quim, Schwarz (or Schwarz-Bayesian) etc

4) In the analysis and conclusion, it is inferred that the accuracy of the results are enhanced with a larger forecast horizon. So how poorly does the model behave for a "small" forecast horizon and indeed, what is large enough for the forecast horizon

The manuscript is written intelligibly well, in standard English and with minimal editorial corrctions

Reviewer #3: 1. “When using the Mixed Data Sampling (MIDAS) regression model to forecast GDP, determining the mixed frequency lag structures using traditional lag structure selection methods reduces forecast accuracy”- Modify this line. Abstract should not start with when.

2. Author must constructively change the abstract in terms of adding only best numerical values to the result. The Author needs to write in a consistent way. Also author mention error analysis of result.

3. Please modify the objective section for a clear understanding i.e novelty part should be clearly mentioned.

4. Author must mention where they got data and frequency of data. Statistical analysis of data must be given in Tabular format (Like Table no 1, 10.1061/(ASCE)IR.1943-4774.0001689, not for citation purposes, only for checking Table 1 ).

5. Please add more recent literature (2024) in terms of the PSO-MV-MIDAS Model for better understanding.

6. Author must add study flowchart, which will clearly specify the research done in this study.

7. Comparison statement (compare with other research articles) must be added in the result and discussion section to better visualize the proposed research.

8. Author must add future scope in the last portion of the manuscript.

9. Advantages and limitations of the proposed model must be added.

10. For better analysis of the result author must add a histogram plot

11. Author must provide a flow chart, pseudo code of proposed individual models.

6. PLOS authors have the option to publish the peer review history of their article (what does this mean?). If published, this will include your full peer review and any attached files.

Reviewer #1: **Yes: **Han Liu

Reviewer #2: **Yes: **Muhammed Bashir Mu'azu

Reviewer #3: No

---

## [Author Response · Author response to Decision Letter 0]

14 Aug 2024

We have studied reviewer’s comments carefully and have made revision which marked in red in the paper. We have tried our best to revise our manuscript according to the comments. Attached please find the revised version, which we would like to submit for your kind consideration.

---

## [Decision Letter · Decision Letter 1]

3 Sep 2024

PONE-D-24-22830R1A novel hybrid PSO-MIDAS model and its application to the U.S. GDP forecastPLOS ONE

Dear Dr. Shen,

Thank you for submitting your manuscript to PLOS ONE. After careful consideration, we feel that it has merit but does not fully meet PLOS ONE’s publication criteria as it currently stands. Therefore, we invite you to submit a revised version of the manuscript that addresses the points raised during the review process.

**ACADEMIC EDITOR: **

Revised manuscript has still some concerns to be dealt with. So, the decision is Minor Revision.

We look forward to receiving your revised manuscript.

Kind regards,

Sercan Ergün

Academic Editor

PLOS ONE

Journal Requirements:

Additional Editor Comments:

Revised manuscript has still some concerns to be dealt with. So, the decision is Minor Revision.

Reviewers' comments:

Reviewer's Responses to Questions

**Comments to the Author**

1. If the authors have adequately addressed your comments raised in a previous round of review and you feel that this manuscript is now acceptable for publication, you may indicate that here to bypass the “Comments to the Author” section, enter your conflict of interest statement in the “Confidential to Editor” section, and submit your "Accept" recommendation.

Reviewer #4: All comments have been addressed

Reviewer #5: All comments have been addressed

2. Is the manuscript technically sound, and do the data support the conclusions?

Reviewer #4: Partly

Reviewer #5: Yes

3. Has the statistical analysis been performed appropriately and rigorously? 

Reviewer #4: Yes

Reviewer #5: Yes

4. Have the authors made all data underlying the findings in their manuscript fully available?

Reviewer #4: Yes

Reviewer #5: Yes

5. Is the manuscript presented in an intelligible fashion and written in standard English?

Reviewer #4: Yes

Reviewer #5: Yes

6. Review Comments to the Author

Reviewer #4: The manuscript presents a well-structured and methodologically sound study that contributes to the field of economic forecasting.

the manuscript could benefit from additional discussions on the practical implications of the findings and a more detailed exploration of the PSO algorithm’s optimization process.

1- Expand the discussion on the implications of the findings for economic policy and forecasting practices.

Reviewer #5: The authors have revised the manuscript properly by incorporating all of the corrections/suggestion, therefore, it is recommended for publication in its present form.

7. PLOS authors have the option to publish the peer review history of their article (what does this mean?). If published, this will include your full peer review and any attached files.

Reviewer #4: **Yes: **Hadi Abbasian

Reviewer #5: No

---

## [Author Response · Author response to Decision Letter 1]

20 Sep 2024

Considering the Reviewer’s suggestion, we have incorporated suggestions for implementing the PSO-MIDAS model in the conclusion section (see line 489 in the revised manuscript).

---

## [Decision Letter · Decision Letter 2]

29 Oct 2024

PONE-D-24-22830R2A novel hybrid PSO-MIDAS model and its application to the U.S. GDP forecastPLOS ONE

Dear Dr. Shen,

Thank you for submitting your manuscript to PLOS ONE. After careful consideration, we feel that it has merit but does not fully meet PLOS ONE’s publication criteria as it currently stands. Therefore, we invite you to submit a revised version of the manuscript that addresses the points raised during the review process.

**ACADEMIC EDITOR: **The manuscript has critical concerns to deal with. 

We look forward to receiving your revised manuscript.

Kind regards,

Sercan Ergün

Academic Editor

PLOS ONE

Journal Requirements:

Additional Editor Comments (if provided):

The manuscript has critical concerns to deal with.

Reviewers' comments:

Reviewer's Responses to Questions

**Comments to the Author**

1. If the authors have adequately addressed your comments raised in a previous round of review and you feel that this manuscript is now acceptable for publication, you may indicate that here to bypass the “Comments to the Author” section, enter your conflict of interest statement in the “Confidential to Editor” section, and submit your "Accept" recommendation.

Reviewer #6: (No Response)

2. Is the manuscript technically sound, and do the data support the conclusions?

Reviewer #6: Yes

3. Has the statistical analysis been performed appropriately and rigorously? 

Reviewer #6: Yes

4. Have the authors made all data underlying the findings in their manuscript fully available?

Reviewer #6: No

5. Is the manuscript presented in an intelligible fashion and written in standard English?

Reviewer #6: Yes

6. Review Comments to the Author

Reviewer #6: Feng Shen et. al. have presented a Particle Swarm Optimization algorithm to determine the lag parameters for contributing factors in GDP forecasting using the MIDAS model. The manuscript is well presented and the analysis is robust.

There are a few further improvements that I would like to suggest to add clarity, and improve understanding:

1. It will be helpful to include temporal plots of actual vs forecasted GDP, and to derive conclusions about the models’ ability to forecast positive vs negative growth, since RMSE does not account for sign.

2. Line 80 - Sentence is unclear.

3. Line 98 - Chapter -> Section

4. Line 118 and Abstract - Could you please clarify the metric which is improved by 10%? The abstract mentions forecast accuracy, which could be further clarified as X-month forecast accuracy.

5. Eq 1 - The term u_{t+1} is undefined. It is not clear why there is a future time term on the right hand side of the equation. Same for Eq 2, 6.

6. Eq 2 - Does the equation make an assumption that the high frequency data X^{M} is monthly? This also applies to later parts of the manuscript and would be helpful to clarify.

7. Line 233 - It would be helpful to describe “large” as a typical number for the parameters to be estimated. For example, 100 vs 10,000.

8. Line 234 - Please expand and describe RMSFE at first usage.

9. Line 246 - Is the PSO algorithm capable of always finding the global optimum, or does it simply perform better than other optimization algorithms in finding the global solution? The sentence is confusing.

10. Line 265 - What is the purpose of r1 and r2 in the equation? Is this how randomness is introduced in the algorithm?

11. Line 295 - It is not clear why the lag parameters are continuous. Wouldn’t they be discrete - 0, 1, 2, 3, …?

12. Line 400 - The meaning of optimization effect is unclear.

13. Line 401 - The summary numbers in the text - 37%, 25%, 15% - are hard to interpret from Table 3. It would be helpful to add more explanations behind calculating these results. Also applies to later results.

14. Line 414 - Please expand and describe MAPE at first usage.

15. Line 419 - Please describe the Diebold and Mariano test.

16. Line 486 - It is subjective to state that the model was significantly better given the variability in performance. I would suggest removing “significantly”.

17. Line 492 - It would be helpful to include the units of forecast horizon, such as, 2 quarters.

18. Line 491 - Is this conclusion correct for each of the three variables in the univariate model?

7. PLOS authors have the option to publish the peer review history of their article (what does this mean?). If published, this will include your full peer review and any attached files.

Reviewer #6: No

---

## [Author Response · Author response to Decision Letter 2]

24 Nov 2024

We have responded to the reviewers’ questions in document “Response to Reviewers.docx” and describe the changes made to the manuscript.

---

## [Editor Report · Decision Letter 3]

28 Nov 2024

A novel hybrid PSO-MIDAS model and its application to the U.S. GDP forecast

PONE-D-24-22830R3

Dear Dr. Shen,

We’re pleased to inform you that your manuscript has been judged scientifically suitable for publication and will be formally accepted for publication once it meets all outstanding technical requirements.

Kind regards,

Sercan Ergün

Academic Editor

PLOS ONE
---

## [Editor Report · Acceptance letter]

1 Dec 2024

PONE-D-24-22830R3 

PLOS ONE

Dear Dr. Shen, 

I'm pleased to inform you that your manuscript has been deemed suitable for publication in PLOS ONE. Congratulations! Your manuscript is now being handed over to our production team.

Kind regards, 

on behalf of

Dr. Sercan Ergün 

Academic Editor

PLOS ONE